# Investigation of Biomechanical Differences in Level Walking between Patients with Bilateral and Unilateral Total Knee Replacements

**DOI:** 10.3390/bioengineering11080763

**Published:** 2024-07-28

**Authors:** Derek Yocum, Alejandro Ovispo-Martinez, Kevin A. Valenzuela, Chen Wen, Harold Cates, Songning Zhang

**Affiliations:** 1South Bend Orthopaedics, South Bend, IN 46635, USA; 2Department of Kinesiology, Recreation, & Sport Studies, The University of Tennessee, Knoxville, TN 37996, USA; 3Department of Kinesiology, California State University Long Beach, Long Beach, CA 90840, USA; 4Tennessee Orthopedic Clinic, Knoxville, TN 37922, USA

**Keywords:** arthroplasty, gait, knee, hip, ankle

## Abstract

Due to the high risk of a bilateral total knee arthroplasty (TKR) following unilateral TKR, this study was performed to investigate bilateral TKR patients. Specifically, we examined biomechanical differences between the first replaced and second replaced limbs of bilateral patients. Furthermore, we examined bilateral TKR effects on hip, knee, and ankle biomechanics, compared to the replaced and non-replaced limbs of unilateral patients. Eleven bilateral patients (70.09 ± 5.41 years, 1.71 ± 0.08 m, 91.78 ± 13.00 kg) and fifteen unilateral TKR patients (65.67 ± 6.18 years, 1.73 ± 0.10 m, 87.72 ± 15.70 kg) were analyzed while performing level walking. A repeated measures one-way ANOVA was performed to analyze between-limb differences within the bilateral TKR group. A 2 × 2 (limb × group) ANOVA was used to determine differences between bilateral and unilateral patients. Our results showed that the second replaced limb exhibited a lower peak initial-stance knee extension moment than the first replaced limb. No other kinematic or kinetic differences were found. Bilateral patients exhibited lower initial-stance knee extension moments, knee abduction moments, and dorsiflexion moments, compared to unilateral patients. Bilateral patients also exhibited lower push-off peak hip flexion moments and vertical GRF. The differences between the first and second replaced limbs of bilateral patients may indicate different adaptation strategies used following a second TKR. The significant group differences indicate that adaptations are different between these groups, and it is not recommended to use patients with unilateral and bilateral TKR together in gait analyses.

## 1. Introduction

The majority of the estimated 8.6 million people with severe knee OA in the United States are likely to receive a total knee replacement (TKR). It is expected that by 2030, 3.5 million TKR surgeries will be performed annually [1]. Between 37% and 46% of unilateral TKR patients will undergo TKR in the contralateral limb within 20 years [2,3,4]. Given the trend of increasing total TKR surgeries and the high percentage of TKR patients eventually needing their contralateral limb replaced, it is important that we understand how the presence of bilateral knee replacements alters lower extremity gait biomechanics. Limited research has examined the level-walking biomechanics of bilateral TKR patients [5,6,7,8]. Borden et al. [8] found that staged bilateral TKR patients had a lower knee flexion range of motion (ROM) and peak initial-stance (IT) vertical GRF than asymptomatic controls and simultaneous TKR patients. Bolanos et al. [7] found no differences in peak knee extension moments and ROM between cruciate-retaining and cruciate-substituting implants in patients who had undergone simultaneous (one-staged) TKR. Renaud et al. [5] compared the kinematics of two different types of cruciate-substituting TKR implants and found that the second replaced limb had less adduction ROM from the initial contact to midstance. No joint kinetics were reported. It is currently unknown if the significant time between replacements causes altered biomechanics between the first and second replaced limbs. Such an investigation may be beneficial, as advancements in TKR designs and degradation of older TKRs may generate differences between these limbs, which may have implications in the quality of life and rehabilitation strategies for those patients.

The presence of an implant may produce altered joint kinematics and kinetics in the remaining joints (i.e., hip and ankle) in the lower limb of TKR patients. Two studies have examined how knee replacement affects hip and ankle kinematics and kinetics [9,10]. Levinger et al. [9] found no differences in the hip joint kinematics or kinetics between unilateral TKR patients and asymptomatic controls. However, higher peak dorsiflexion angles were found in the replaced limb of the unilateral patients. No kinetic differences were identified at the ankle. These results, however, conflict with the recent study by Biggs et al. [10], who found increased hip flexion angles, reduced hip adduction ROM, reduced peak hip external flexion moments, and a loss of the biphasic nature of the hip adduction moment, compared to the asymptomatic controls. The unilateral TKR patients exhibited increased dorsiflexion and ankle internal rotation moments during the first half of stance, and lower dorsiflexion and internal rotation moments during the second half of stance. To our knowledge, no studies have examined differences in the hip, knee, and ankle biomechanics between the first and second replaced limbs of bilateral TKR patients in gait.

Pain level, knee society score, and physical functions have been examined in the bilateral and unilateral TKA patients, and have been shown to be either similar [11] or more superior for the bilateral group [12]. However, no studies have compared the hip, knee, and ankle biomechanics of bilateral TKR patients to the replaced and non-replaced limbs of unilateral TKR patients. Previous research on TKR patients excluded bilateral TKR patients [13,14] or included bilateral TKR patients within their patient population without examining interlimb differences in bilateral patients [15]. If joint mechanics of the first and second replaced limbs of bilateral TKR patients differ from those of unilateral TKR patients, it may not be a good idea to include both bilateral and unilateral patients in the same gait biomechanics study. The primary purpose of this study was to examine differences in knee joint biomechanics in both limbs of bilateral TKR patients and the replaced and non-replaced limbs of unilateral TKR patients during level walking. Our first hypothesis is that bilateral TKR patients would have similar knee extension and abduction moments and ROM between the first and second replaced limbs. Our second hypothesis was that bilateral TKR patients would exhibit similar knee extension and abduction moments and ROM compared to the replaced limb of unilateral TKR patients, but decreased knee extension moments and ROM compared to the non-replaced limb of unilateral TKR patients.

## 2. Materials and Methods

### 2.1. Participants

For this study, 15 (6 males) bilateral TKR patients (69.23 ± 5.23 years, 1.73 ± 0.09 m, 95.56 ± 15.24 kg) were recruited from a local orthopedic clinic. Eleven of the patients had staged bilateral replacements (73.36 ± 21.92 months since the first TKR and 59.00 ± 25.11 months since the second TKR). We tested all 15 participants but excluded 4 of them in the analyses because they had undergone simultaneous bilateral knee replacement. Additionally, fifteen (eight male) unilateral TKR patients (65.67 ± 6.18 years, 1.73 ± 0.10 m, 87.72 ± 15.70 kg, 27.93 ± 12.03 months since TKR) were randomly selected from two previous studies conducted in our lab [16,17]. The inclusion criteria for all patients included men and women between 50 and 75 years of age, at least 12 months from most recent TKR, no more than 10 years since first TKR, cruciate retaining implant, and surgeries performed by the same surgeon. The exclusion criteria were OA in the hip or ankle, any additional lower extremity joint replacement, a BMI greater than 40, neurological disease, and inability to walk or negotiate stairs without the use of a walking aid or handrail. An a priori power analysis was performed using a previous report of knee extension moments in bilateral TKR patients, compared to healthy controls [6]. This analysis indicated a minimum of 15 participants was needed per group to achieve a beta of 0.8 and an alpha of 0.05. All participants signed an informed consent document, and all procedures were approved by the Institutional Review Board at The University of Tennessee, Knoxville.

### 2.2. Instrumentation

Three-dimensional (3D) kinematics were collected using a 12-camera motion analysis system (240 Hz, Vicon Motion Analysis Inc., Oxford, UK). All participants wore standardized running shoes. Anatomical markers were placed bilaterally on the first and fifth metatarsal heads, distal end of the second toes, medial and lateral malleoli and femoral epicondyles, greater trochanters, iliac crests, and acromion processes. Semi-ridged thermoplastic shells, each with four retroreflective markers, were used for motion tracking. These shells were placed bilaterally on the lateral shanks and thighs, on the dorsal aspect of each midfoot, as well as on the distal posterior trunk. Furthermore, the pelvis was tracked using a pair of shells, each with two retroreflective markers, placed on the posterior pelvis. The hip joint center was calculated at 25% of the distance between greater trochanters [18]. A force platform (1200 Hz, BP600600, American Mechanical Technology Inc., Watertown, MA, USA) measured 3D ground reaction forces (GRF) and moments. Two sets of photocells (63501 IR, Lafayette Instrument Inc., IN, USA) and electronic timers (54035A, Lafayette Instrument Inc., IN, USA), set three meters apart, monitored the time for the participants to complete each trial.

### 2.3. Experimental Procedures

Upon arrival, participants were asked to fill out forms to assess their capacity to perform physical activity, pain, functional capacity, and satisfaction. The survey used to collect this data was the Physical Activity Readiness Questionnaire. Unilateral patients from one of the previous studies [17] completed the Western Ontario and McMaster Universities Osteoarthritis Index [19]. Participants then proceeded to warm up at a self-selected pace on a treadmill for three minutes.

Participants were allowed up to five practice trials to familiarize themselves with the over-ground walking conditions. Average walking speed (±10%) was determined from practice trials and used to moderate data collection trials to ensure that the gait speed of actual test trials fell within the speed range for consistency. Participants performed 3–5 data collection trials for each of the level walking conditions.

The testing conditions included walking with the foot of the first TKR limb contacting the force platform and with the foot of the second TKR limb contacting the force platform, respectively. The condition order (first/second TKR) was randomized for all patients. Trials were repeated if the incorrect foot was used to step on the force platform, if the foot was outside the boundaries of the force platform, if the participant altered their gait to actively target the force platform, or if the predetermined speed range was not achieved.

### 2.4. Data Analysis

3D kinematics and kinetic computations were performed using Visual 3D biomechanical software suite (Version 6.0, C-Motion, Inc., Germantown, MD, USA). A fourth-order Butterworth low-pass filter was used to filter raw marker and GRF data at cutoff frequencies of 8 Hz and 50 Hz, respectively. Furthermore, 3D angular kinematic and kinetic computations were performed using a Cardan rotational sequence (X-Y-Z), and conventions were defined using the right-hand rule. Positive values of the ankle, knee, and hip indicate dorsiflexion, inversion, internal rotation, knee/hip adduction, knee extension, and hip flexion. Joint moments were calculated as internal moments. Customized computer programs (VB_V3D and VB_Table, MS Visual Basic 6.0, Redmond, WA, USA) were used to identify and organize critical values and events. The events and critical values correspond with phases within the stance phase. The IT phase is 0–25% of the stance phase and the push-off (PO) phase is defined as 75–100% of the stance phase. These critical values were averaged across 3–5 trials for each condition and phase to be used for the statistical comparison. Joint moments and the GRF values were normalized to the participant mass (Nm/kg) and body weight (BW), respectively.

### 2.5. Statistical Analysis

To test our first hypothesis, a repeated measure one-way analysis of variance (ANOVA) was performed to identify kinematic and kinetic differences between first and second replaced limbs of bilateral TKR patients. A 2 × 2 (limb × group) mixed-model ANOVA was performed to detect kinematic and kinetic differences between bilateral and unilateral TKR groups. Both ANOVA tests had an alpha level of 0.05 set a priori. All statistical tests were performed using IBM SPSS Statistics (version 24). The observed power of the main effects and interactions was reported as partial eta squared (η_p_^2^). Post hoc comparisons were performed on significant interactions using a stepwise Holm adjustment for multiple comparisons [20,21]. The effect size of Cohen’s d was calculated and reported for all *t*-tests and considered as small (d = 0.2), medium (d = 0.5), or large (d ≥ 0.8).

## 3. Results

Bilateral TKR patients recruited for this study were similar in age, height, and weight to the unilateral patients (Table 1). The average time since the first TKR of bilateral TKR patients was 10.5 months earlier than the second TKR (*p* = 0.003, d = 1.30). Furthermore, times since surgery for the first replaced limb (*p* < 0.001, d = 2.44), and second replaced limb (*p* < 0.001, d = 1.69), of bilateral patients was significantly longer than the replaced knee of the unilateral patients (Table 1). There were no differences in walking speed between groups (Table 1).

During the IT of the stance phase, no limb (*p* = 0.133) or group (*p* = 0.195) main effect differences were found for vertical GRF (Table 2). During the PO of the stance phase, a significant group GRF main effect, showing decreased vertical GRF in bilateral TKR patients, was found (F(1, 28) = 6.63, *p* = 0.016, η_p_^2^ = 0.191).

At the knee, a significant within-group difference was found, indicating the first replaced limb of bilateral patients had a significantly higher IT peak knee extension moment, compared to the second replaced limb (*p* = 0.024, d = 0.925, Table 2 and Figure 1). Furthermore, a limb × group interaction (F(1, 27) = 5.76, *p* = 0.024, η_p_^2^ = 0.176) was found for the peak IT extension moment. Post hoc tests demonstrated that the peak moment for the first replaced (*p* = 0.010, d = −1.03) and second replaced (*p* < 0.001, d = −1.60) limbs of bilateral patients were significantly lower than non-replaced limbs of unilateral patients. Furthermore, the peak knee extension moment for the second replaced limb was lower than for unilateral replaced limbs (*p* = 0.001, d = −1.44). A main effect of group was also identified for the IT abduction moment (F(1, 28) = 5.04, *p* = 0.033, η_p_^2^ = 0.153). This main effect demonstrates that the bilateral group had lower IT peak knee abduction moments than the unilateral group.

At the ankle, the peak IT dorsiflexion moment was significantly higher in the unilateral group (F(1, 28) = 18.24, *p* < 0.001, η_p_^2^ = 0.394 Table 2). No further significant group/limb main effects, or interactions, were found at the ankle.

The hip joint also exhibited kinetic differences in the sagittal plane. The PO flexion moment was significantly lower in the bilateral group (F(1, 28) = 7.78, *p* = 0.009, η_p_^2^ = 0.217). A significant interaction in the hip adduction ROM was identified (F(1, 28) = 4.25, *p* = 0.049, η_p_^2^ = 0.132, Table 3). No significant post hoc comparisons were found.

## 4. Discussion

The primary purpose of this study was to examine the differences in the knee joint biomechanics of both limbs of bilateral and unilateral TKR patients during level walking. Our first hypothesis was that the first and second knee replacements of bilateral TKR patients would exhibit similar peak knee extension and abduction moments, as well as ROM. This hypothesis was partially supported. The first replaced limb of bilateral patients exhibited significantly higher peak initial-stance knee extension moments, which disagrees with previous research that has shown bilateral TKR patients exhibiting similar peak knee extension moments in each limb [7,8]. When comparing the motions of both first and second replaced knees, we found no differences in knee extension and abduction ROM throughout the stance phase. This is a positive outcome as it suggests that both knees exhibit similar joint kinematic patterns during gait. However, the decreased peak knee extension moment in the second replaced limb may indicate a more complex recovery following a second replacement. The quadriceps avoidance gait, commonly associated with reductions in the knee extension moment, seems to be more prevalent in the second replacement limb of bilateral TKR patients. Silvia et al. [22] determined that reduced quadriceps strength was present in the replaced limb of unilateral patients 2.8 years after surgery, compared to asymptomatic controls. Huang et al. [23] found that reduced quadriceps strength persisted up to 13 years following surgery. Our results also showed reduced initial-stance peak knee extension moment for the first replaced limb compared to the replaced limb of unilateral TKR patients, suggesting that these patients may have increased difficulties recovering quadriceps strength in the second replaced limb.

Our second hypothesis, that bilateral TKR patients would have similar peak knee extension and abduction moments and knee joint ROM during the stance phase as in the replaced limb of unilateral TKR patients was partially supported. The peak initial-stance knee extension moment was significantly lower in the bilateral TKR group. Post hoc analysis indicated lower moments in both limbs of bilateral TKR patients, compared to the non-replaced limb of unilateral TKR patients. This was expected, as it is a similar finding as previous research on bilateral patients and asymptomatic controls [6]. Unlike the study by Ro et al. [6], our bilateral patients did not walk significantly slower than the opposing group. Despite similar walking speeds, a lower extension moment continued to persist in bilateral patients. This may indicate increased movement efficiency in the bilateral group, allowing for reduced moments while maintaining their walking speed. Furthermore, the second replaced limb of bilateral patients was significantly lower than the replaced limb of unilateral patients.

This decreased moment may be representative of a quadriceps avoidance that is more prevalent in the bilateral group. Similar to the previous discussion on the differences of this variable within the bilateral group, the presence of two knee replacements may cause an exaggerated quadriceps avoidance gait in bilateral patients. Furthermore, non-significant differences between these groups may have helped to promote this difference. The bilateral patients walked slightly slower and had a slightly lower initial-stance peak vertical GRF, which may have collectively contributed to decreased initial-stance peak knee extension moments [24]. Despite this reduction in the initial-stance moment, the push-off knee flexion moment was slightly (but non-significantly) higher in the bilateral group. Similarities in knee extension and abduction ROM between these groups indicate that both groups use similar knee kinematic movement patterns during level walking. The clinical implications of these results suggest that although the patients with bilateral TKR may appear to be normal in their gait as shown with their “normal” kinematics, the second replaced limbs appear weaker in their knee extensors during level walking. Rehabilitation may focus more on that limb, both in terms of strength training and gait retraining efforts.

Additionally, the unilateral group demonstrated no between-limb differences for the initial-stance peak knee extension moment. This contrasts with previous research, which has demonstrated lower peak knee extension moments in the replaced limbs of unilateral patients, compared to their non-replaced limbs, as well as asymptomatic controls [9,15,25,26]. A lack of differences in the initial-stance peak knee extension moment may be associated with the use of contralateral limbs of unilateral TKR patients. A recent study by Aljehani et. al. [27] found there were differences between the limbs of unilateral TKR patients which depended on the presence of bilateral OA. This research found that the patients with bilateral OA had symmetrical, abnormal joint motions following unilateral TKR. However, patients who were asymptomatic in the non-operated knee had asymmetrical joint motions, with increased initial contact knee flexion, less knee flexion and extension excursion, and decreased knee extension in the non-replaced limb, compared to replaced. Therefore, their results suggested that contralateral limbs of TKR patients may not be as useful for comparisons as asymptomatic controls.

Furthermore, the initial-stance internal knee abduction moment (KAbM) was lower in the bilateral group. This was an expected result, as previous research comparing replaced and non-replaced limbs of unilateral TKR patients has shown the initial-stance KAbM to be significantly lower in the replaced limb, compared to non-replaced limbs [28,29,30]. This peak has also been found to be smaller in the replaced limb of unilateral TKR patients compared to asymptomatic controls [30,31,32]. Reduced KAbM is a positive sign, as increased KAbM is commonly associated with increased loading on the medial compartment of the knee [28,30]. A previous study found that bilateral TKR patients may achieve higher functional scores than unilateral patients [12]. Furthermore, reduced KAbM is supported by previous research on between-limb differences in unilateral patients. There is conflicting evidence for the differences between asymptomatic controls and non-operated limbs. Alnahdi et al. [30] found no differences between control and non-replaced limbs. Milner and O’Bryan [28] found no difference between replaced and control limbs, but the non-replaced limb was higher than both limbs. In our study, we found no differences in KAbM between the replaced and non-replaced limbs of unilateral patients. This is similar to a recent study by Wen et al. [16] who found no difference in KAbM between the replaced and non-replaced limbs of unilateral patients during level walking.

In addition to examining the knee joint, our secondary purpose was to examine any differences between or within these groups at the ankle and hip. We hypothesized that bilateral patients would have similar hip and ankle kinematics and kinetics between first and second replaced limbs, which was supported, as ankle dorsiflexion/plantarflexion moments and hip extension/flexion and abduction moments, as well as ankle dorsiflexion/eversion and hip extension/adduction ROM, were similar between the limbs of the bilateral patients. A lack of differences between limbs indicates that these patients may have developed similar neuromuscular adaptations in both first and second replaced limbs. This result also reflects the similar movement patterns at the knee. Furthermore, the slightly (non-significant) higher initial-stance hip extension moment in the second replaced limb may be present to compensate for the initial-stance knee extension moment in the second replaced limb.

Additional between-group differences were identified at the hip and ankle. During IT, the bilateral patients exhibited reduced dorsiflexion moments compared to the unilateral patients. Reduced dorsiflexion moments in bilateral patients were not expected. A recent study by Biggs et al. [10] found that the replaced limb of TKR patients had higher peak dorsiflexion moments, compared to asymptomatic controls. It was theorized that unilateral patients might have relied on increased dorsiflexion moments to compensate for muscle weakness at the knee joint. This indicates different compensation methods between these groups at the ankle. Biggs et al. [10] also found that the replaced limb of unilateral patients had reduced peak hip external flexion moments, compared to asymptomatic controls. This contrasts with Levinger et al. [9], who found no differences in hip kinetics. Furthermore, our study did not find different initial-stance peak hip extension moments in bilateral patients compared to unilateral patients. However, bilateral patients exhibited lower push-off hip flexion moments, as well as push-off peak vertical GRF. These lower joint moments may be related to the high functional scores, and, therefore, higher functional capacity, of our bilateral patients, compared to the unilateral patients.

Limitations of this study include a longer time since surgery for bilateral patients, compared to unilateral patients. These scores may not be representative of all bilateral TKR patients. This was a cross-sectional study, and we cannot determine the longitudinal effects of bilateral surgery on gait biomechanics. A longitudinal study measuring the gait patterns over time should be conducted in future studies in this population. Secondly, a high body weight of TKR patients may produce soft tissue motion artifacts during level gait, which are not representative of the underlying skeletal movements [33,34]. Thirdly, the small sample size of the bilateral TKR group was a limitation of this study. We would like to also recognize the importance of other factors, such as biomaterials and viscosupplements [35] in the success of TKR implants. Future studies should consider incorporating these aspects related to TKR biomechanics and quality of life.

## 5. Conclusions

Our results demonstrated that the bilateral group had a lower initial-stance knee extension moment in the second replaced limb. Bilateral patients also exhibited significantly lower KAbM, initial-stance dorsiflexion moment, and push-off hip flexion moment compared to the unilateral patients. Aside from these differences, bilateral patients had similar loading response and push-off hip, knee, and ankle joint sagittal- and frontal-plane joint moments, as well as ROM compared to the unilateral patients, and between the first and second replaced limbs. These results indicate that the bilateral patient population may produce neuromuscular adaptations that are different than those of unilateral patients. Due to significant differences between bilateral and unilateral patients, it is not recommended to use bilateral TKR patients in conjunction with unilateral patients when examining their gait biomechanical adaptations. Future research on how acute adaptations differ following first and second replacements may be needed to understand why these groups differ. Previous research has shown no difference in pain level and functional scores 30 days after unilateral and bilateral TKR surgeries. Furthermore, research into more physically demanding daily activities, such as stair negotiation and ramp negotiation, may be warranted to examine how these patients differ.

## Figures and Tables

**Figure 1 bioengineering-11-00763-f001:**
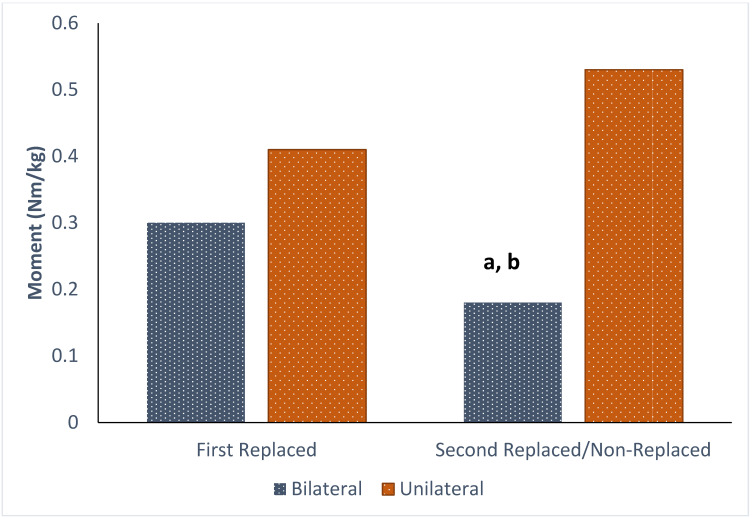
Peak knee extension moments for the bilateral and unilateral patient groups. **^a^** Significantly different from unilateral replaced limb, and **^b^** significantly different from unilateral non-replaced limb.

**Table 1 bioengineering-11-00763-t001:** Patient demographics: mean ± STD.

	Bilateral	Unilateral	*p*
Age (years)	70.09 ± 5.41	68.67 ± 6.18	0.547
Height (m)	1.71 ± 0.08	1.73 ± 0.10	0.524
Weight (kg)	91.78 ± 13.00	87.73 ± 15.70	0.493
Time since First TKR (months)	73.36 ± 21.92	27.93 ± 12.03	**<0.001**
Time since Last TKR (months)	59.00 ± 25.11	27.93 ± 12.03	**0.002**
Walking speed (m/s)	1.07 ± 0.13	1.18 ± 0.21	0.079

Bold: significant difference.

**Table 2 bioengineering-11-00763-t002:** Peak GRFs (N/kg) and ankle, knee, and hip moments (Nm/kg): mean ± STD.

Variable	Bilateral		Unilateral		Limb *p*	Group *p*	Int. *p*
	First Replaced	Second Replaced	Replaced	Non-Replaced			
IT Vertical GRF	1.04 ± 0.13	1.05 ± 0.12	1.07 ± 0.06	1.09 ± 0.04	0.280	0.299	0.676
PO Vertical GRF	1.00 ± 0.06	1.00 ± 0.07	1.05 ± 0.06	1.06 ± 0.05	0.207	**0.026**	0.456
IT Knee Ext. Moment	0.30 ± 0.26	0.18 ± 0.21 ^a,b^	0.41 ± 0.14	0.53 ± 0.26	0.925	**0.008**	**0.034**
PO Knee Flex. Moment	−0.12 ± 0.09	−0.14 ± 0.15	−0.08 ± 0.14	−0.14 ± 0.13	0.183	0.620	0.494
IT Knee Abd. Moment	−0.34 ± 0.07	−0.38 ± 0.12	−0.46 ± 0.10	−0.45 ± 0.18	0.613	**0.037**	0.404
PO Knee Abd. Moment	−0.28 ± 0.12	−0.30 ± 0.12	−0.32 ± 0.08	−0.36 ± 0.16	0.358	0.260	0.723
IT DF Moment	0.16 ± 0.04	0.15 ± 0.04	0.30 ± 0.11	0.27 ± 0.07	**0.049**	**<0.001**	0.281
PO PF Moment	−1.28 ± 0.11	−1.21 ± 0.27	−1.30 ± 0.16	−1.34 ± 0.15	0.717	0.238	0.110
IT Hip Ext. Moment	−0.62 ± 0.20	−0.67 ± 0.19	−0.59 ± 0.13	−0.55 ± 0.14	0.799	0.201	0.158
PO Hip Flex. Moment	0.47 ± 0.11	0.43 ± 0.13	0.61 ± 0.17	0.62 ± 0.17	0.561	**0.006**	0.395
IT Hip Abd. Moment	−0.87 ± 0.10	−0.90 ± 0.15	−0.91 ± 0.13	−0.92 ± 0.19	0.584	0.521	0.687
PO Hip Abd. Moment	−0.84 ± 0.13	−0.88 ± 0.18	−0.85 ± 0.11	−0.83 ± 0.14	0.597	0.708	0.242

^a^ Significantly different from unilateral replaced following Holm adjustment, ^b^ Significantly different from Unilateral Non-Replaced following Holm Adjustment, IT: initial-stance, PO: push-off, PF: plantarflexion, DF: dorsiflexion, Int.: limb × group Interaction, bold: significant *p*-values.

**Table 3 bioengineering-11-00763-t003:** Ankle, knee, and hip ROM (deg): mean ± STD.

Variable	Bilateral		Unilateral		Limb *p*	Group *p*	Int. *p*
	First Replaced	Second Replaced	Replaced	Non-Replaced			
Knee Extension	−47.42 ± 4.63	−47.93 ± 5.73	−46.06 ± 5.80	−47.15 ± 6.26	0.472	0.594	0.797
Knee Abduction	3.74 ± 2.07	3.33 ± 1.71	4.11 ± 1.24	3.52 ± 0.83	0.127	0.652	0.786
Ankle Dorsiflexion	22.80 ± 3.62	23.69 ± 4.84	24.70 ± 3.03	22.55 ± 3.67	0.396	0.772	**0.049**
Ankle Eversion	−8.06 ± 4.50	−7.19 ± 3.70	−5.96 ± 2.24	−7.23 ± 3.63	0.836	0.319	0.275
Hip Extension	−34.00 ± 5.72	−32.85 ± 5.32	−34.33 ± 6.08	−37.18 ± 4.79	0.402	0.466	0.056
Hip Adduction	9.16 ± 2.57	7.59 ± 4.15	11.56 ± 4.16	10.45 ± 4.98	**0.044**	0.097	0.721

Int.: Limb × group interaction, bold: significant *p*-values.

## Data Availability

The data presented in this study are available on request from the corresponding author due to privacy restrictions.

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
