# Peer review of "Investigation of Biomechanical Differences in Level Walking between Patients with Bilateral and Unilateral Total Knee Replacements"

_bioengineering, 2024, doi:10.3390/bioengineering11080763_

Round 1

Reviewer 1 Report

Comments and Suggestions for Authors

This is a well-written article with a nice structure. The materials and methods are designed appropriately and stated clearly. The reviewers only have two suggestions:

(1) Please highlight the clinical significance and possible applications of the findings in the Discussion section.

(2) There are just a few typos. Please carefully read the article and correct all typos.

Author Response

Thank you for your valuable comments.

Reviewer 1

This is a well-written article with a nice structure. The materials and methods are designed appropriately and stated clearly. The reviewers only have two suggestions:

Comment (1) Please highlight the clinical significance and possible applications of the findings in the Discussion section.

Response 1: We have added some discussions on the clinical implications of our primary results in the discussion (lines 258 – 262).

Comment (2) There are just a few typos. Please carefully read the article and correct all typos.

Response 2: We have gone through a thorough proof reading and corrected a number of typos and grammatic issues.

Reviewer 2 Report

Comments and Suggestions for Authors

Dear Author,

Thank you for the opportunity to review this article. It covers an interesting topic about whether it is reliable to assess level walking among unilateral and bilateral TKR patients altogether. However:

• Upon a first reading, we advise a moderate English revision. We recommend an expert translator or a program such as Grammarly for this process. • In the Introduction, you might want to include general information about quality of life among unilateral and bilateral TKR patients, and the implications of kinematics without punctual references that may be better mentioned in the Discussion. • Materials and methods are thorough. • The Results are clear, although a graphic illustration of the gait analysis could be useful. • The conclusions should comprise 3-7 clear phrases based on your research. Please reformulate.

Author Response

Thank you for your valuable comments.

Reviewer 2

Dear Author,

Thank you for the opportunity to review this article. It covers an interesting topic about whether it is reliable to assess level walking among unilateral and bilateral TKR patients altogether.

However:

  • Upon a first reading, we advise a moderate English revision. We recommend an expert translator or a program such as Grammarly for this process.

Thank you for the comment. We have made many adjustments to the wording and grammar in the revised manuscript.

  • In the Introduction, you might want to include general information about quality of life among unilateral and bilateral TKR patients, and the implications of kinematics without punctual references that may be better mentioned in the Discussion.

We added references of studies of quality-of-life related research for the bilateral and unilateral TKR patients in the introduction (Lines 68-70).  We also added the potential benefits of this research on quality of life and rehabilitation strategies for this patient group in the introduction (Lines 52-53).

As to moving some of implications of kinematics to the discussion, we feel that these discussions are necessary to provide relevant background information in the introduction. The implications of the kinematic related discussions can be found in the discussion.

  • Materials and methods are thorough.

Thank you.

  • The Results are clear, although a graphic illustration of the gait analysis could be useful.

We added Figure 1 to demonstrate the outcome of one of the main variable, peak knee extension moment.

  • The conclusions should comprise 3-7 clear phrases based on your research. Please reformulate.

Thank for your recommendation. We have reformatted the conclusion and left out a few recommendations for future research.

Reviewer 3 Report

Comments and Suggestions for Authors

This is a very good and timely original article on biomechanics differences between bilateral and unilateral total knee replacement patients. The manuscript fits the journal scope, the quality is adequate, although there are minor changes to be addressed prior proceeding to publication.  First of all, the title should be changed: according to the iThenticate report, it is exactly the same as retrieved over the first author thesis content. From a scientific soundness point of view, it is anyway very tricky and not appealing. The abstract section shall be in a manuscript-form, with a consistent flow, without needing of mention all sections titles ("Background, Methods, Results, Conclusions" fits for a PhD thesis, not for an article format). Considering the strong use of abbreviations throughout the manuscript, a list of abbreviation will provide for both boosting the manuscript and help the reader. Moreover, although interesting, the manuscript completely avoid an entire branch of approaches, such as biomaterials, viscosupplements and so forth. This can be retrieved in a poor reference appropriateness, which requires to be addressed accordingly. Therefore, as mentioned in the appropriate section, it is also suggested to improve and update the introduction and references, mainly for what concerns about side approaches, such as biomaterials and so forth (e.g. DOI: 10.1016/j.ijbiomac.2017.04.079).
An English editing is strongly required: e.g. "inadvisable"?

It is also suggested to add on a "future perspectives" few sentences, either as a separate section, after the conclusions section, or include it over the conclusions.

Comments on the Quality of English Language

An English editing is strongly required: e.g. "inadvisable"?

Author Response

Thank you for your valuable comments.

Reviewer 3

This is a very good and timely original article on biomechanics differences between bilateral and unilateral total knee replacement patients. The manuscript fits the journal scope, the quality is adequate, although there are minor changes to be addressed prior proceeding to publication. 

First of all, the title should be changed: according to the iThenticate report, it is exactly the same as retrieved over the first author thesis content. From a scientific soundness point of view, it is anyway very tricky and not appealing.

We have changed the title to “Investigation of Biomechanical differences of level walking between patients with bilateral and unilateral total knee replacements”. To clarify, the dissertation’s title is different from the original title of this manuscript.

The abstract section shall be in a manuscript-form, with a consistent flow, without needing of mention all sections titles ("Background, Methods, Results, Conclusions" fits for a PhD thesis, not for an article format).

We have revised the format of the abstract to remove the section titles.

Considering the strong use of abbreviations throughout the manuscript, a list of abbreviation will provide for both boosting the manuscript and help the reader.

We have added the list of abbreviations at the end of the manuscript.  We also removed a couple of the abbreviations which are not used more than once.

Moreover, although interesting, the manuscript completely avoid an entire branch of approaches, such as biomaterials, viscosupplements and so forth. This can be retrieved in a poor reference appropriateness, which requires to be addressed accordingly. Therefore, as mentioned in the appropriate section, it is also suggested to improve and update the introduction and references, mainly for what concerns about side approaches, such as biomaterials and so forth (e.g. DOI: 10.1016/j.ijbiomac.2017.04.079).

Thank you for raising these important and interesting aspects of TKR implants. We added a statement in the discussion section about the importance of these aspects for future studies related to TKR (Lines 328 – 331). Due to the nature of our study which is less directly related to these factors, we choose not to include a discussion in the introduction,

An English editing is strongly required: e.g. "inadvisable"?

Thank you for your comments and we have done a thorough English check throughout the manuscript including changing “inadvisable” to “not be recommended”.

It is also suggested to add on a "future perspectives" few sentences, either as a separate section, after the conclusions section, or include it over the conclusions.

We moved and added some future perspectives at the end of the conclusion (Lines 343 - 348).
